# CdSSe Nano-Flowers for Ultrasensitive Raman Detection of Antibiotics

**DOI:** 10.3390/molecules28072980

**Published:** 2023-03-27

**Authors:** Kangkang Wang, You Li, Honggang Wang, Ziyue Qian, Xiaokai Zhu, Sabir Hussain, Liming Xie

**Affiliations:** 1CAS Key Laboratory of Standardization and Measurement for Nanotechnology, National Center for Nanoscience and Technology, Beijing 100190, China; 2School of Nanoscience and Technology, University of Chinese Academy of Sciences, Beijing 100049, China

**Keywords:** nano-flower, surface-enhanced Raman scattering, enrichment, antibiotics

## Abstract

Surface-enhanced Raman scattering (SERS) technique is widely used for the highly sensitive detection of trace residues due to its unparalleled signal amplification ability and plays an important role in food safety, environmental monitoring, etc. Herein, CdSSe nano-flowers (CdSSe NFs) are synthesized via the chemical vapor deposition (CVD) method. CdSSe NFs thin film is used as a SERS substrate with an ultralow limit of detection (LOD, 10^−14^ M), high apparent enhancement factor (EF, 3.62 × 10^9^), and excellent SERS stability (relative standard deviation, RSD = 3.05%) for probe molecules of Rh6G. Further, CdSSe NFs substrate is successfully applied in the sensitive, quantitative, and label-free analysis of ciprofloxacin (CIP) and enrofloxacin (ENR) antibiotics, which exhibit LODs of below 0.5 ppb. This excellent SERS platform may be widely utilized for sensitive life science and environmental sensing.

## 1. Introduction

Antibiotics have been used for disease treatment and growth regulation in crops and animals [1]. For example, enrofloxacin (ENR), a fluoroquinolone antibiotic, has been widely used for preventive and therapeutic purposes in farm animals and fishery products [2]. Ciprofloxacin (CIP), the primary metabolite of ENR, is one of the most used antimicrobials in feed additives [3]. However, residual CIP and ENR antibiotics can cause water pollution, affect the ecological environment, as well as be absorbed into the human body through the food chain, which can lead to the emergence of antibiotic resistance [4,5]. Detection of antibiotics is very important in agriculture, food processing, and human health [6]. So far, the sensitive detection of antibiotics at trace levels is highly desired but far from satisfactory [7,8]. Chromatography and enzyme immunoassays provide rapid and sensitive detection of antibiotics [9,10]. The capillary electrophoresis technique can be applied to the detection of a wider range of residues [11]. The click chemistry method is also successfully applied in the selective determination of antibiotics [12]. For example, Chen et al. have used an electrochemiluminescence immunosensor for the detection of ENR with a detection limit of 0.001 ppb [4]. Xu et al. have reported capillary electrophoresis with field amplified sample stacking-sweeping technique to achieve the sensitive determination of CIP and ENR with a low limit of detection (LOD) below 2.5 ppb [13]. In addition, high-performance thin-layer chromatographic and fluorescence-linked immunosorbent methods are also viable strategies for detecting CIP and ENR [14,15]. However, the experimental procedures of the above detection methods are complicated. Therefore, it is particularly important to find a rapid, accurate, and highly sensitive method to achieve the detection of antibiotics [16,17].

Surface-enhanced Raman scattering (SERS) is a promising technique for sensitive and quantitative analysis of residues at the ultralow concentration, benefiting from its extraordinary signal amplification capacity [18,19]. Recently, SERS-based sensors have been successfully applied to trace detection of drugs, pesticide residues, and antibiotics [20,21,22,23]. Initially, precious metals (Au, Ag) were used as the SERS substrates with enhancement mechanisms mainly derived from the surface plasmon resonance (SPR) effect [24]. However, the scarcity of precious metals significantly limits their commercial applications.

Conversely, semiconductor-based SERS substrates represented by two-dimensional (2D) transition metal dichalcogenides (TMDs) are emerging as new SERS substrates [25]. The SERS performance of 2D materials (such as graphene and MX_2_ (M = Mo, W, V, Nb, Ta; X = O, S, Se, Te)) is mainly contributed by the chemical mechanism (CM) of Raman enhancement and the fluorescence quenching [19]. Specifically, the photo-induced CT process can directly affect the electron density distribution or polarizability of the analyte molecules and finally magnify the Raman intensity [26,27]. So far, an increasing number of semiconductor materials are successively designed as SERS substrates, enabling accurate identification of heavy metal ions, antibiotics, uric acid, and other residues at low concentrations [25]. It is worth mentioning that semiconductor nanowires (NWs) have significant quantum effects, large specific surface area, and good prospects for applications in optoelectronic devices, sensors, and SERS applications [28,29,30]. Substrates composed of NWs can form rich hot spots due to the electric field enhancement contribution of individual nanowires, allowing them to exhibit excellent SERS performance [31]. Nanowire-based SERS substrates have been reported mainly for plasma-based metal systems [32].

Here, substrates fabricated by CdSSe NFs exhibited excellent SERS activity and reproducibility, which enabled ultrasensitive recognition of rhodamine 6G (Rh6G) and methyl blue (MB) probe molecules with an ultralow LOD and a high apparent enhancement factor (EF). The enhancement mechanism of the CdSSe NFs-based SERS system could be attributed to the enrichment effect and light-induced CT between the analyte and the substrate surface. Moreover, the substrate was also successfully applied to the trace, quantitative, and label-free analysis of CIP and ENR, which are commonly used in farming and meat processing in the food industry [33]. CdSSe NFs thin film with sensitive SERS response may further pioneer more applications in food safety detection.

## 2. Results and Discussion

### 2.1. Characterizations of CdSSe NWs and NFs

CdSSe NWs and NFs were obtained by chemical vapor deposition (CVD) deposition on Si substrate using Au film as a catalyst (Figure 1a). The growth mechanisms of NWs and NFs could be attributed to vapor-liquid-solid (VLS) growth and Plateau-Rayleigh crystal growth, respectively [34,35]. The SEM images of CdSSe NWs could be observed in Appendix A. CdSSe NWs exhibited a staggered distribution of structure with a length of about 50 µm and a diameter of about 500 nm. As shown in Figure 1b,c, CdSSe NFs (about 6 µm) with a lot of nanowires accumulated in a cluster-like structure and closely connected together. As displayed in Figure 1d, the energy dispersive spectroscopic (EDS) mapping images showed that CdSSe NFs consisted of Cd, S, and Se elements and were distributed uniformly. Figure 1e showed the X-ray diffraction (XRD) analysis, and most of the typical diffraction peaks of CdSSe NFs were coincident with CdSe and CdS reactants. There was also the appearance of new peaks and changes in peak position, indicating the formation of ternary compounds. As shown in Figure 1f, the Raman spectrum of CdSSe NFs appeared the characteristic peaks of CdSe (52, 96, and 206 cm^−1^) and CdS (302 cm^−1^) powders, which is consistent with the previous reports [36,37]. As we all know, nanostructures and bulk powder are spatially different in size, so the slight variation in the peak position of CdSSe NFs may be due to the phonon confinement caused by size effects [38]. Additionally, the X-ray photoelectron spectroscopy (XPS) technique was applied to further determine the composition of products. The wide-scan survey XPS spectrum of CdSSe NFs could be observed in Appendix A. Apart from the C and O elements, peaks including 11.4 eV (Cd 4d3), 404.4 eV (Cd 3d), and 617.1 eV (Cd 3p3) were assigned to the Cd element. Peaks at 54.9 eV (Se 3d) and 162.6 eV (S 2p) were assigned to the Se and S elements, respectively. The high-resolution XPS spectra of Cd 3d, S 2p and Se 3d were illustrated in Figure 1g–i. Concretely, the strong peaks located at around 411.9 and 405.2 eV belonged to Cd 3d_3/2_ and Cd 3d_5/2_, respectively. Peaks appeared on 163.4 and 161.6 eV, corresponding to S 2p_1/2_ and S 2p_3/2_. And peaks at around 54.9 and 54.1 eV were assigned to Se 3d_3/2_ and Se 3d_5/2_, respectively [39,40].

### 2.2. SERS Property of CdSSe NFs

As shown in Figure 2a, CdSSe NFs/H_2_O dispersions and analytes were deposited onto a SiO_2_/Si substrate in turn by spin-coating method to fabricate the CdSSe NFs-based system. Rh6G and MB were selected as the probes to verify the SERS activity of the CdSSe NFs substrate, and the detailed Raman test conditions were available in the Method part. Figure 2b,c show that CdSSe NFs had better SERS activity than CdSSe NWs for Rh6G and MB probe molecules, which may be due to a higher enrichment of the adsorbed probe molecules in CdSSe NFs. Therefore, CdSSe NFs-based substrates were eventually used for SERS experiments. As exhibited in Figure 2d, the SERS signals of Rh6G at different concentrations were highly consistent with expanding the ultralow LOD to 10^−14^ M, which suggested a concentration-dependent SERS phenomenon. Surprisingly, some typical peaks of Rh6G, including 612 cm^−1^ (C−C−C ring in-plane bending), 774 cm^−1^ (C−H out of plane bending), 1184 cm^−1^ (C−H in-plane bending), 1362 cm^−1^ (CH_3_ bending), 1507 cm^−1^ (β(C−H)(CH_3_), β(C−H)(ring)), 1573 cm^−1^ (C−C stretching vibrations), and 1650 cm^−1^ (C−C stretching vibrations), can be clearly detected even at the low concentration of 10^−14^ M (insert on the top left of Figure 2d) [41]. Such a strong SERS response exhibited by CdSSe NFs is surprising, which is close to the performance of most metal-based SERS substrates. In order to visually observe the SERS effect of the substrate on the Rh6G probe, we performed a quantitative statistical analysis of the peak intensity of Rh6G at varying concentrations. Figure 2e shows the trend of peak intensity with concentration, indicating the ability of the CdSSe NFs substrate in trace detection. As illustrated in Figure 2f, the apparent EFs of Rh6G increased sharply with decreasing concentration. After calculation, the maximum apparent EF values at 1184 cm^−1^ (R_1_), 1362 cm^−1^ (R_2_), 1573 cm^−1^ (R_3_), and 1650 cm^−1^ (R_4_) were 3.62 × 10^9^, 1.82 × 10^9^, 1.84 × 10^9^, and 1.79 × 10^9^ respectively.

Another MB probe was also tested to confirm the SERS universality of the CdSSe NFs-based platform. Appendix A indicated that CdSSe NFs-based substrate also had good SERS activity for the MB probe with a low LOD of 10^−13^ M and a high apparent EF (2.12 × 10^9^). Obviously, the SERS activity of CdSSe NFs to MB was slightly lower than that of Rh6G, owing to the selectivity of CdSSe NFs for the analyte [42]. In fact, MB molecules at a low concentration may cause partial photolysis under the laser, which could also affect its detection limit [43]. Therefore, the CdSSe NFs substrate could reach the ultrasensitive recognition of probe molecules at trace levels comparable to metal-based SERS substrates. Besides, the apparent EFs of Raman peaks of Rh6G and MB probes at different concentrations can be observed in Appendix A in detail. Currently, the SERS mechanism of the semiconductor-based substrate is mainly attributed to fluorescence quenching and light-induced CT between the analyte and substrate surface. The CT processes in most semiconductors usually occur at the valence and conduction band edges of the substrate material [44]. In addition, the SEM images of CdSSe NFs suggested that these nano-flowers are clustered structures formed by the accumulation of multiple nanowires. The enhancement mechanism of CdSSe NFs may also originate from the enrichment effect of the substrate on probe molecules (Figure 3). Firstly, the surface of CdSSe nano-flowers has abundant contact sites, which can sufficiently adsorb the probe molecules. Then, the well-developed pores inside the nano-flowers could generate an enrichment effect on probes. Thus, the Raman signals of probes at low concentrations can be easily detected using CdSSe NFs-based substrate. Thus, we consider that the enhancement mechanism of CdSSe NFs originates from the photo-induced CT process and enrichment of probes molecules by CdSSe NFs thin film.

### 2.3. SERS Reproducibility and Uniformity

For a substrate, reproducibility and uniformity are also two important indicators to be considered, except for the remarkable SERS activity [45]. In this work, 30 random SERS spectra of the Rh6G probe (10^−6^ M) on CdSSe NFs substrate were collected to evaluate the stability of the substrate during the testing process. As reflected in Figure 4a, these spectral signals showed good agreement in both Raman shifts and intensities, suggesting good reproducibility of spectra in the CdSSe NFs system. Figure 4b, the contour map of Figure 4a, showed that the above SERS spectra have almost identical spectral patterns compared to the average SERS spectrum (white line). Going further, the relative standard deviations (RSDs) values of peak intensities were also calculated. Figure 4c,d shows that the RSDs of P_1_ (1650 cm^−1^) and P_2_ (1362 cm^−1^) were about 3.05% and 3.84%, respectively, within a reasonable and acceptable limit. These results obviously demonstrate that CdSSe NFs-based substrate has a sensitive SERS activity and excellent stability.

### 2.4. SERS Detection of Antibiotics

The accurate and sensitive detection of antibiotic residues at low concentrations facilitates the regulation and prevention of antibiotic abuse. Here, CdSSe NFs substrates with sensitive SERS response and good stability were finally applied to the trace detection of CIP and ENR. Incredibly, the CdSSe NFs platform could accurately identify CIP, and ENR molecules ranged from 10^−3^ M to 10^−9^ M (Figure 5a,c). Peaks at 760, 789, 1353, 1372, and 1620 cm^−1^ of CIP and 638, 1248, 1395, 1465, 1536, and 1626 cm^−1^ of ENR could be observed clearly even when the concentration decreased to 10^−9^ M [46]. Liu et al. used 2D niobium ditelluride (NbTe_2_) nanosheets as SERS substrates to realize the detection of CIP and ENR at low concentrations (about 35 ppb) [22]. In this work, such low LODs levels of 0.351 ppb (CIP) and 0.359 ppb (ENR) were equivalent to silver nanoparticle-based substrates [47]. Furthermore, a good linear dependence (R^2^ of 0.99) between the SERS intensity at 1372 cm^−1^ and the logarithmic concentrations of CIP was obtained (Figure 5b). Accordingly, such a good linear relationship was also reflected in the SERS detection of ENR (Figure 5d). CdSSe NFs as nonmetal substrates presented here provided a reliable strategy for quantitative SERS analysis of antibiotics. More importantly, SERS technology could be used for multicomponent detection benefiting from the narrow-band feature of the Raman signal [48]. Figure 6 shows the capability of CdSSe NFs substrate in the detection of CIP and ENR mixture (10^−7^ M). The fingerprint peaks of CIP and ENR could be markedly distinguished from the SERS spectrum of the mixture. Hence, CdSSe NFs-based SERS platform could be reliable in trace, quantitative, and label-free analysis of antibiotics in water samples, indicating the practical implications for preventing the abuse of antibiotics.

## 3. Materials and Methods

### 3.1. Materials

CdS powder (99.999%), CdSe powder (99.995%), rhodamine 6G (Rh6G), methyl blue (MB), and absolute ethyl alcohol were purchased from Aladdin Reagents (Shanghai, China). Ciprofloxacin (CIP) and enrofloxacin (ENR) were obtained from Acmec Biochemical Co., Ltd. (Shanghai, China). All reagents were used without further purification.

### 3.2. Preparation of CdSSe NWs and CdSSe NFs

CdSSe NWs and CdSSe NFs were synthesized by a simple one-step CVD method using Au film as a catalyst at ambient pressure. In brief, the mixture of the CdS and CdSe powders in the molar ratio of 1:1 was placed in a ceramic boat in the middle of a horizontal single-zone furnace. The silicon substrates coated with 5 nm gold film were placed downstream about 18 cm away from the source powders. Prior to heating, high-purity N_2_ was introduced into the quartz tube with a constant flowing rate (50 sccm) to purge the O_2_ inside, then fast heating to 1000 °C within 30 min and keeping at this temperature for 60 min with the gas (Ar (95%)/H_2_ (5%)) flow of 20 sccm. When the furnace temperature is naturally cooled to room temperature, CdSSe nanowires with vertically interlaced distribution are deposited on the substrate. The CdSSe nano-flowers could be obtained by moving the substrate closer to the downstream insulation and extending the annealing time to 120 min.

### 3.3. Characterization

The SEM images were acquired using a Hitachi SU8220 microscope equipped with an energy-dispersive spectroscopic (EDS) microanalysis system (OXFORD). XRD patterns were recorded using D/Max-TTRIII (CBO) with Cu Kα radiation (λ = 1.54056 Å). Raman spectra were collected using a Renishaw inVia plus spectrometer with a 514 nm laser. XPS measurements were conducted using an ESCALAB 250Xi electron spectrometer from VG Scientific with 300 W Al Kα radiation.

### 3.4. Preparation of CdSSe NFs SERS Substrate and Raman Measurement

CdSSe NFs SERS substrate was obtained by the spin-coating method. Specifically, the SiO_2_/Si substrate was washed with acetone, ethanol, and deionized water in sequence. Simply, 5 μL of CdSSe NFs dispersion was dribbled onto the surface of SiO_2_/Si and formed a dense thin film after spinning. Then, 5 μL of probes with different concentrations (Rh6G: 10^−4^ M to 10^−14^ M, MB: 10^−4^ M to 10^−13^ M) were consecutively spin-coated onto the surface of the above substrate. The laser power was 0.5 mW with a 5 s exposure time under the 514 nm laser excitation. Cosmic ray removal, smoothing, and baseline correction were done for all Raman spectra using Wire 4.4 software. Specifically, the baseline correction was done using the ‘Intelligent fitting’ with a polynomial order of 11 and a noise tolerance of 1.5. Furthermore, the SERS intensities of some typical Raman peaks were calculated to quantitatively describe the Raman enhancement. We calculated the apparent enhancement factors (AEF) values using the following Equation [49]:$$\mathrm{AEF}=\left(I_{\mathrm{SERS}}{\text{·}C}_{\mathrm{Raman}}\right)/\left(I_{\mathrm{Raman}}{\text{·}C}_{\mathrm{SERS}}\right)$$
where *I*_SERS_ and *C*_SERS_ refer to the Raman intensity and concentration of probes in the SERS spectra, and *I*_Raman_ and *C*_Raman_ are the Raman intensity and concentration of probes in the Raman spectra. For example, 5 μL of pure Rh6G and MB probes (10^−3^ M) were spin-coated onto the SiO_2_/Si substrate for testing to obtain *I*_Raman_ values. In Appendix A, the Raman peaks selected for the calculation of AEF values of SERS spectra at different concentrations (including the lowest observed concentration) are the same (Rh6G: 1184, 1362, 1573 and 1650 cm^−1^; MB: 1177, 1365, 1578 and 1618 cm^−1^). And we use the average of the peak intensities at different concentrations as the approximate values to obtain *I*_Raman_ and *I*_SERS_ values.

## 4. Conclusions

Altogether, we have synthesized CdSSe NFs and first used them for the fabrication of SERS substrates. CdSSe NFs thin film exhibited excellent SERS performance and stability, which was comparable to that of metal-based SERS substrates. The SERS mechanism can be attributed to the enrichment effect of the substrate on probe molecules and the photo-induced CT process that occurs between the substrate and analyte. In addition, this substrate was also successfully applied in the sensitive, quantitative, and label-free recognition of CIP and ENR antibiotics at ppb levels. Thus, the ultrasensitive SERS platform exerts great flexibility, which may be further applied to food safety and environmental protection.

## Figures and Tables

**Figure 1 molecules-28-02980-f001:**
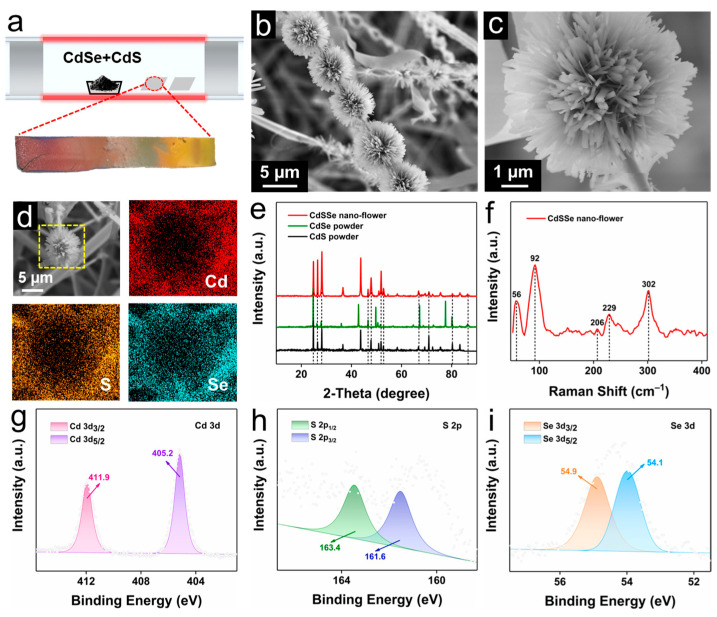
(**a**) Schematic representation of the experimental setup for the synthesis of CdSSe NWs and NFs. (**b**,**c**) SEM images of CdSSe NFs at different magnifications. (**d**) The elemental mapping images of CdSSe NFs (the yellow box is the selected test area). (**e**) XRD patterns of CdSSe NFs, CdSe and CdS powders. (**f**) Raman spectra of CdSSe NFs. (**g**–**i**) High-resolution XPS spectra of Cd 3d, S 2p, and Se 3d in CdSSe NFs.

**Figure 2 molecules-28-02980-f002:**
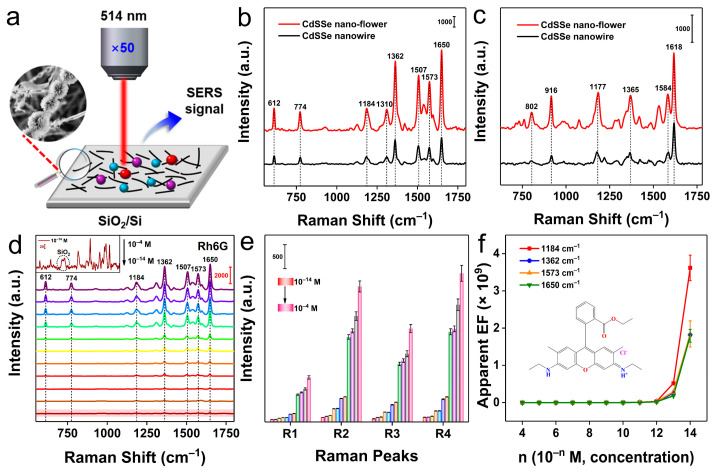
(**a**) Schematic illustration of CdSSe NFs-based substrate SERS experiments. (**b**,**c**) Comparison of the SERS performance of CdSSe NWs and CdSSe NFs substrates for Rh6G and MB probes, respectively. (**d**) SERS spectra of Rh6G solutions at different concentrations (inset shows the SERS spectrum of Rh6G at 10^−14^ M concentration). (**e**) The trend of concentration-dependent Raman peak intensity (R_1_ to R_4_: 1184, 1362, 1573, and 1650 cm^−1^). (**f**) Apparent EF-concentration function in CdSSe NFs-Rh6G SERS systems.

**Figure 3 molecules-28-02980-f003:**
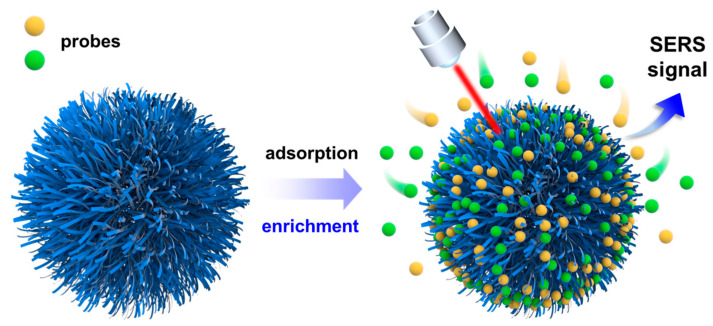
Possible SERS mechanism of CdSSe NFs-based system.

**Figure 4 molecules-28-02980-f004:**
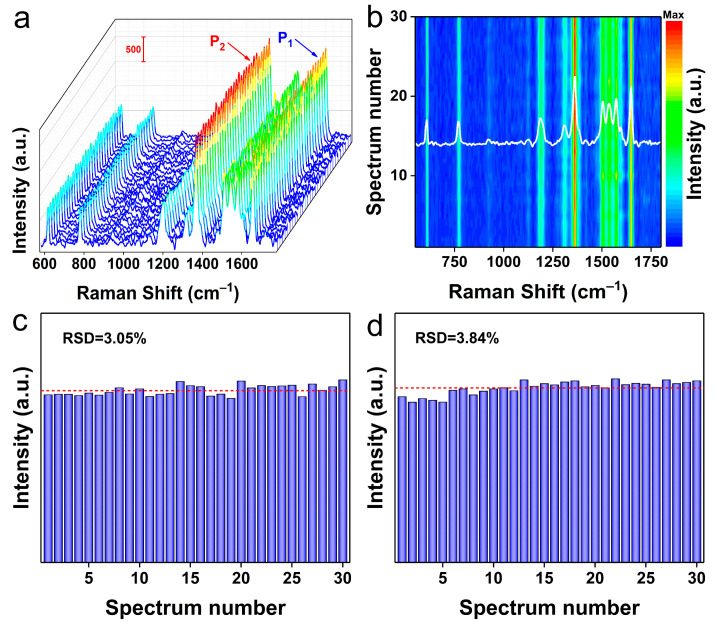
(**a**) Reproducibility of 30 random SERS spectra of Rh6G (10^−6^ M) deposited on CdSSe NFs substrate. (**b**) Relevant mapping patterns of the above SERS spectra. (**c**,**d**) Statistical distribution of peak intensities at P_1_ (1650 cm^−1^) and P_2_ (1362 cm^−1^), respectively.

**Figure 5 molecules-28-02980-f005:**
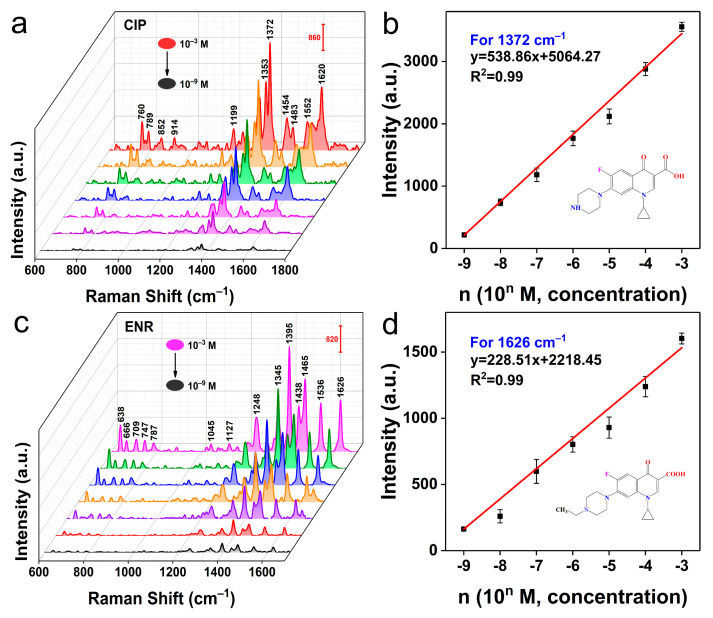
(**a**) SERS spectra of CIP at varying concentrations from 10^−3^ M to 10^−9^ M. (**b**) Fitting curve of SERS intensity-concentration plots for 1372 cm^−1^ of CIP. (**c**) SERS spectra of ENR at varying concentrations from 10^−3^ M to 10^−9^ M. (**d**) Fitting curve of SERS intensity-concentration plots for 1626 cm^−1^ of ENR.

**Figure 6 molecules-28-02980-f006:**
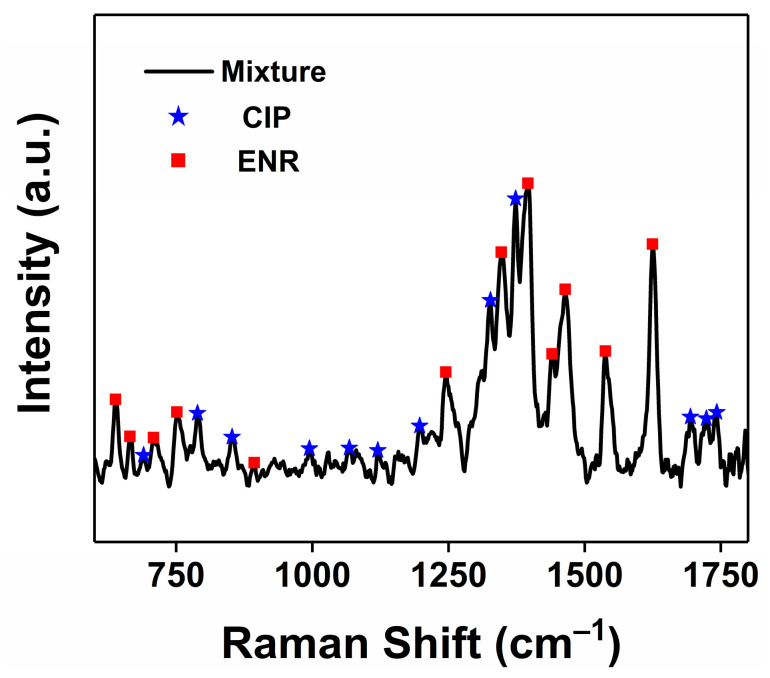
SERS spectra of a mixture of CIP and ENR using the CdSSe NFs as SERS substrate.

## Data Availability

Data available on request from the authors.

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
