# Peer review of "CdSSe Nano-Flowers for Ultrasensitive Raman Detection of Antibiotics"

_molecules, 2023, doi:10.3390/molecules28072980_

Round 1
Reviewer 1 Report
The authors propose the manuscript entitled: CdSSe Nano-flowers for Ultrasensitive Raman Detection of 2 Antibiotics as Communication in the journal: Molecules.
The manuscript is not written very well, the introduction just slides on the cover of knowledge and the insight is not very deep. For example, in the manuscript is not mentioned what is the source of the antibiotics in the substance that the authors would like to investigate. Why the antibiotics are important in agricultures, food process etc. The state of the art connected to the detection method of antibiotics is summarized in two sentences (lines 25-27). The new detection methods are not mentioned et all (for example this article: 10.1002/smll.202207216 ).
The introduction is followed by the section: Results and Discussion section, which begins with a description of the manufacturing process of the SERS structure.
The other SERS substrates are not mentioned et all (even in the introduction part). Why?
In this part are shown very promising results with no discussion, interpretation. For example, on the lines is mentioned that the authors used laser of 514 nm as an excitation laser. Why did you choose this laser wavelength? (It can be also source of fluorescence excitation for a lots of biological media).
The references used for the enhancement factor definition is not original, why? The Origin of the idea of the definition of enhancement factor comes from: 10.1021/jp0687908. In the case of enhancement factor, how do you obtain the value CSERS (line 210)?
The whole text of the manuscript describes very good results but the introduction, interpretation and deep discussion are missing. Because of these reasons, I cannot recommend the manuscript to be accepted as Communication in Molecules.
Reviewer 2 Report
The manuscript "CdSSe Nano-flowers for Ultrasensitive Raman Detection of Antibiotics" shows interesting results on alternatives for its application in SERS. I have an observation about a couple of figures in the manuscript, as it should always be clear to the reader about the interpretation of the results:
1. What is the difference between Figure 4a and Figure 4b?
2. Where do you mention the spectral transformation to image that was carried out?
3. Referring to Figure 4b, he only mentions this text throughout the manuscript: “Figure 4b showed that these spectra have almost identical spectral pattern compared to the average SERS spectrum (white line). the map?, there is a transformation of the Raman intensity values to an image representation (8 bits, 10 bits, etc). You should explain or include it in the methodology part in case you use this type of graph to express the Raman intensity values in pixels, since you place a scale of color bands from minimum to maximum (Max).
Round 2
Reviewer 1 Report
The authors proposed a revision of the manuscript entitled: CdSSe Nano-flowers for Ultrasensitive Raman Detection of Antibiotics.
There has been a lot of improvement in the manuscript. Information that has been added to the introduction helps the reader to follow the line of the experimental story. Despite the considerable improvement of the manuscript, I would give several recommendations to the authors:
1) The level of English needs to be improved. I'm not a native speaker, so I don't feel like judging the level of English but sometimes I have trouble understanding the main idea of a sentence or the sentences are strung together in a strange way. For example:
· Lines 40-41: However, the operation process and stability of the above detection methods are not excellent enough.
· Lines 249: After smoothing and baseline correction, the spectra were reconstructed by Origin Pro 9.1 (Origin Lab, America) software. (Also, I recommend mentioning which baseline you used: polynomial?)
-A peculiar thing to say that: ...., the spectra were reconstructed ...
· lines 255-256: …and concentration of probes in the SERS spectra (I believe that authors mean concentration of the analyte)
2) I recommend to add some references of SERS detection of antibiotics, for example these:
10.1515/nanoph-2013-0024, 10.1007/s00216-016-9957-2, 10.1515/nanoph-2021-0095, 10.1016/j.scitotenv.2019.134956 , but feel free to use different references.
3) For variable at line 254 use single-letter abbreviation something like Aef or something similar.
4) Apparent Enhancement factors calculated in Table S3 show very stable system of enhancement with excelent results. I recommend the authors to mention at the end of the section: „3.4. Preparation of CdSSe NFs SERS Substrate and Raman Measurement“ the values of the enhancement factor (which peaks were used and approximate values with deviations - for the lowest observed concentration).
All these comments and remarks are minor compared to the results obtained by the authors, and I reconsider my decision and recommend that the manuscript be accepted as an article in Molecules after revision and the remarks mentioned above.
Author Response
Please see the attchment
